# New 3D Cone Beam CT Imaging Parameters to Assist the Dentist in Treating Patients with Osteogenesis Imperfecta

**DOI:** 10.3390/healthcare8040546

**Published:** 2020-12-10

**Authors:** Daniela Messineo, Valeria Luzzi, Francesca Pepe, Luca Celli, Arianna Turchetti, Anna Zambrano, Mauro Celli, Antonella Polimeni, Gaetano Ierardo

**Affiliations:** 1Department of Radiological, Oncological and Anatomo-Pathological Sciences, Sapienza University of Rome, 00161 Rome, Italy; 2Department of Oral and Maxillo-Facial Sciences, Sapienza University of Rome, 00161 Rome, Italy; valeria.luzzi@uniroma1.It (V.L.); francesca.pepe13@libero.it (F.P.); antonella.polimeni@uniroma1.it (A.P.); gaetano.ierardo@uniroma1.it (G.I.); 3Rare Bone Metabolism Center, Pediatric Department, Sapienza University of Rome, 00161 Rome, Italy; celli.1585298@studenti.uniroma1.it (L.C.); a.turchetti@policlinicoumberto1.it (A.T.); anna.zambrano@tiscali.it (A.Z.); mauro.celli@uniroma1.it (M.C.)

**Keywords:** CBCT, imperfect osteogenesis, chin, alveolar bone, dentistry

## Abstract

(1) Background: The aim of the work is to identify some imaging parameters in osteogenesis imperfecta to assist the dentist in the diagnosis, planning, and orthodontic treatment of Osteogenesis Imperfecta (OI) using 3D cone beam Computed Tomography (CBCT) and the Double Energy X-ray Absorptiometry (DEXA) technique. (2) Methods: 14 patients (9 males and 5 females; aged mean ± SD 15 ± 1.5) with a clinical-radiological diagnosis of OI were analyzed and divided into mild and moderate to severe forms. The patients’ samples were compared with a control group of 14 patients (8 males and 6 females; aged mean ± SD 15 ± 1.7), free from osteoporotic pathologies. (3) Results: The statistical analysis allowed us to collect four datasets: in the first dataset (C1 sick population vs. C1 healthy population), the t-test showed a *p*-value < 0.0001; in the second dataset (C2 sick population vs. C2 healthy population), the t-test showed a *p*-value < 0.0001; in the third dataset (parameter X of the sick population vs. parameter X of the healthy population), the t-test showed a *p*-value < 0.0001; in the fourth dataset the bone mineralometry (BMD) value detected by the DEXA technique compared to the C2 value of the OI affected population only) the Welch–Satterthwaite test showed a *p*-value < 0.0001. (4) Conclusions: The research has produced specific imaging parameters that assist the dentist in making diagnostic decisions in OI patients. This study shows that patients with OI have a characteristic chin-bearing symphysis, thinned, and narrowed towards the center, configuring it with a constant “hourglass” appearance, not reported so far in the literature by any author.

## 1. Introduction

Osteogenesis Imperfecta (OI), a rare hereditary disease, is a connective tissue disorder characterized mainly by bone fragility; in fact, the affected individuals are particularly predisposed to fractures even following very slight traumas, and the disease is therefore also called “fragile bone syndrome” or “glass bone disease”. According to Sillence’s definition, osteogenesis imperfecta includes a heterogeneous group, both phenotypically and molecularly, of hereditary connective tissue disorders that share similar skeletal anomalies, causing bone fragility and deformity [1,2,3,4] and characterized by osteopenia and the tendency to fracture throughout life [5,6].

The primarily associated factors that help distinguish one type from another include the blue sclera, imperfecta dentinogenesis, early hearing loss (usually conductive), progressive deformity of long bones and spine, and joint hyperclass [1,7,8,9].

The main characteristics of the OI at the craniofacial level are [5,10]: triangular face with convex frontal cone-shaped, class III skeletal pattern, usually by upper jaw retrusion, facial hyperdivergence pattern, tendency to open skeletal bite, thin cranial vault due to the low mineralization, delayed closure of fountains and multiple cranial sutures and Wormian bones (flat supernumerary bones present within the sutures of the skull), and craniocervical junction abnormalities (present in about 37% of patients) [11]. Additionally, OI’s oral manifestations [10,12,13] are very varied, depending on the heterogeneity of the OI phenotype. Dentinogenesis Imperfecta (DI) is the classic oral finding, which divides the OI types of the Sillence classification into two subtypes, depending on the presence or absence of DI. Other oral manifestations are the presence of malocclusions (it is common to find both anterior and posterior open bites, as well as the class III and anterior or posterior crossbites), ectopic eruption and dental agenesis, and periapical radiotransparency [14]. DI, also known as hereditary opalescent dentine, is a hereditary dentine defect that affects both the structure and composition. It is a localized mesodermal dysplasia affecting both primary and permanent dentitions and is considered the most common dental genetic disease. A recent study by Orsini et al. [15] confirmed that there is an altered form of predentin in the deciduous series’ dental elements. On radiographic examination and in cone beam Computed Tomography (CBCT) scans, the crowns look bulbous and show an accentuated cervical constriction, the roots are short, and obliteration of the pulp chamber are present. The four types of OI directly related to collagen mutations (types I–IV) can present DI. It is estimated that DI is present in about half of the patients affected by OI [16,17,18], although not in the same proportion in different types of OI, being more frequent in type III patients, followed by type IV and type I patients. Furthermore, deciduous dentition is often more affected than permanent dentition, and it is estimated that more than 80% of deciduous dentition patients are affected by DI [12]. DI type I is the syndromic form of DI; it is associated with OI. Therefore, its etiology is directly related to that of this disease, associated in the vast majority of cases with mutations of genes coding for type I alpha 1 gene (COL1A1) and collagen type I alpha 2 genes (COL1A2) [19]. DI is often the most penetrating factor of OI [19]. Based on the knowledge of the effect of collagen mutation in fibroblasts, Hall and his collaborators [20] hypothesized that odontoblasts were dysfunctional from the beginning, producing their dilation by intracellular accumulation of abnormal procollagen and degradation products in the raw endoplasmic reticulum, transition vesicles, and secretion vesicles.

DI’s diagnosis is clinical. The enamel is generally normal in thickness and density, but the dental elements may take on a characteristic amber, yellowish-brown, or translucent blue-grey color due to the altered structure of the dentin, which makes them less resistant to caries, wear and more exposed to the risk of enamel fracture [19,21] (Figure 1). An Italian study states that there is no correlation between the degree of dental dyschromia and OI severity [22].

Collagen does not have the same function in bone and dentin, so the severity of bone and dentin involvement is very variable [12,13]. Both teeth are affected by DI, although the deciduous teeth are more severely affected than the permanent ones. In a minority of patients, although the deciduous dentition is clinically affected, the permanent dentition is not, but it is not possible to predict whether permanent dentine may be affected by DI [22].

The differential diagnosis of DI should be made with dentin dysplasia, amelogenesis imperfecta, dental fluorosis, congenital erythropoietic porphyria, and tetracycline staining [19]. Radiographically they are characterized by short and narrow roots, with dentinal hypertrophy leading to obliteration of the pulp, before or immediately after the eruption [23]. The expressiveness of clinical signs is, however, very variable even in the same individual, with some dental elements showing complete obliteration of the pulp, while others show normal dentine.

We must consider that the teeth of patients with OI, apparently without DI symptoms, may show radiographic abnormalities and that teeth showing slight radiographic abnormalities may show a wide range of clinical symptoms. These alterations, although in mild forms, are not very evident, and can also be identified using the CBCT scan study. During a cone beam CT scan, the scanner rotates around the patient’s head in 30 s, providing numerous separate images. The software collects the data and reconstructs the images, generating a digital volume composed of isotropic voxels of the acquired anatomical data, which can then be “reconstructed” with appropriate software in the various planes of interest. An advantageous feature of this equipment is the low dose delivered compared to an investigation with CT and Dentascan reconstruction software now routinely used in dentistry. One of the disadvantages of this technique is that small patients can create motion artifacts if they do not cooperate. In addition, imaging diagnostics uses OI patient diagnostics integrated with computerized bone mineralometry (BMD); the measurement has proven to be a useful tool to assess the progress of the disease and/or the effect of treatment in individual patients [24]. Double Energy X-ray Absorptiometry (DEXA), using photonic absorption at two different energy levels, overcomes the problem of soft tissue surrounding the bone allowing the measuring of the bone mineral density in different skeletal districts: lumbar spine, femur, and skeleton. If the bone density is lower than normal, it indicates a risk of osteoporosis and bone fractures which in OI patients is a frequent occurrence. The clinical practice standard is to obtain a DEXA scan before beginning treatment with bisphosphonates. The patient scanning, in the supine position, is carried out according to a system of Cartesian axes and the computer reconstructs in pixels. Through a software, the operator selects the area of interest in the explored field and the equipment provides the values of bone mineral content (BMC) and bone mineral density (BMD); the latter is calculated by dividing the BMC by the area and is expressed in g/cm^2^. The BMD value is expressed in terms of standard deviations (BMD of the subject under examination—average BMD of the reference population/ standard deviation (DS) of the reference population) and is called the Z score. The values obtained are reported on a reference curve, which represents the normality curve. Patients with OI may have a lower than average Z score and BMD score, but the children’s Z score is not reliable because the young patients are still growing. In small patients with OI, it is therefore common to assess the BMD value.

The same would happen with histological and microscopic results, so that apparently normal teeth on clinical examination may present structural abnormalities examined under the microscope.

The objectives of early treatment in deciduous dentition are [25]: maintaining oral health and preserving the vitality, shape, and size of teeth, providing the patient with as pleasant an aesthetic appearance as possible, especially at a young age, in order to prevent psychological problems, ensure functional teeth, prevent loss of vertical dimension, and maintain arch length, avoiding interference with permanent teething eruption allowing normal growth of facial bones and temporomandibular joint (TMJ), establishing a relationship of trust with the patient and their family at an early stage. In patients with DI, it is, therefore, essential to carry out close dental and orthodontic monitoring from the first year of life.

The objective of this work has been to identify some imaging parameters that assist the dentist in the diagnosis, planning, and orthodontic treatment of OI. The primary objective remains to implement a protocol of professional oral hygiene and conservative therapies, to try to minimize wear and tear, and to prevent caries processes from reaching the pulp to avoid infectious complications such as pulpits and abscesses. Orthodontic intervention to prevent malocclusions due to the loss of dental elements or part of them is also significant. This method of study is the first of its kind; there are no similar studies in the literature and no one had ever noticed the possibility of correlating the physiognomic variation of the chin with an objective parameter that can relate to the clinical severity of the disease.

## 2. Materials and Methods

For this study, 14 patients (9 males and 5 females; mean age ± SD: 15 ± 1.5) with clinical-radiological and genetics diagnosis of OI divided into mild and moderate-severe forms were selected in the Pediatric Dentistry Unit of ‘’Policlinico Umberto I’’ Hospital, Sapienza University of Rome, underwent consultation at the Rare Bone Metabolism Center, Pediatric Department Sapienza University of Rome. The population sample was compared with a control population of 14 patients (8 males and 6 females; aged mean ± SD 15 ± 1.7) free from osteoporotic pathologies. All patients signed written informed consent forms, and the study was approved by the Institutional Review Board of territorial National Health Service facilities (n. 260919).

The instrumental investigations examined were Cone Beam Computed Tomography (CBCT), for the oral and maxillofacial regions and dual X-ray absorptiometry (DEXA) technique, for the study of the lumbosacral column. The patient population underwent CBCT investigation, requiring an evaluation of the second phase of orthodontic therapy, concerning a possible treatment of fixed multibracket orthodontics. CBCT scans were performed with a NewTom V Gi Dental X-ray Machine (QR, Verona, Italy) with the following technical data: field of view 12 × 8, kV 110, exposure time 5.3 s, total mA 64.68, delivered dose 7.7 mGy. We performed evaluations of imaging parameters for the morphology of the chin morphology. The dental arch we considered for CBCT was the lower dental arch. Patients with chin or skull syndromic malformations were excluded from the study. Patients with OI in different clinical forms, from mild (a) to moderate (b) to severe (c) and population control, were examined (Table 1 and Table 2).

In our retrospective analysis, we identified the following parameters and adhered to strict reference planes for the measurement with CBCT. On the paraxial plane, elements 3.1 and 4.1 were identified, and measurements were made on these elements; in fact, from this line, the eminence of the chin guard symphysis (X) was evaluated orthogonally (90°) (Figure 2a). On the coronal plane, positioning between the two lower central incisors, the maximum point (C1), and the minimum point (C2) of the chin guard symphysis cortical on the horizontal plane were evaluated (Figure 2b).

Lumbar spine bone mineral density (BMD) was assessed by means of a DEXA scan (Discovery-A, Hologic). Additionally, in the group of affected patients, bone densitometry (DEXA) was performed, and in particular, the parameter BMD (g/cm^2^) was correlated with parameter C2—i.e., the amplitude of the average minimum chin-lowering symphysis. Bone density was measured using dual energy X-ray absorptiometry (DEXA) (Hologic Horizon 4500) with pediatric software analysis.

### Statistical Analysis

The results were expressed as absolute values by taking measurements in centimeters. For both populations surveyed, data from CBCT examinations were measured. The data of the two populations were considered independent and extracted from two populations at unknown variance. The data were analyzed with the following tests: F-test or Fisher test and Student t-test. Fisher’s F-test allowed verification of the homogeneity of the variances, obtaining a *p*-value < 0.05, with an index of non-homogeneity between samples. Then, a t-test was performed on the two samples with the calculation of the degrees of freedom according to the Welch–Satterthwaite formula, which resulted in a *p*-value < 0.05, indicating a statistically significant difference between the two populations. A value of *p* < 0.05 was set as an index of significance. Data from the study of the two populations were analyzed using GraphPad Prism 8 Software (San Diego, CA, USA).

The statistical analysis was, therefore, organized by entering four sets of data: the first dataset compared the amplitude of the maximum coronal chin-lowering symphysis (C1) of the sick group to the healthy one; the second set of data compared the amplitude of the minimum coronal chin-lowering symphysis (C2) of the sick group to the healthy group; the third dataset compared parameter X of the population affected by OI to parameter X of the population of the control group; the fourth dataset compared the Bone Mineral Density (BMD) value detected by the DEXA technique to the C2 value in the OI population only.

## 3. Results

The examined cohort of the sick population had an average age of 15 years (±1.5 DS), while the control population has an average age of 15 (±1.7 DS). The measurements in centimeters measured in CBCT and DEXA (gr/cm^2^) have been reported in Table 1 for patients with OI. Table 2 shows C1 and C2 data for the control group. In the first dataset (C1 sick population vs. C1 healthy population), the t-test showed a *p*-value of < 0.0001 (Figure 3); in the second dataset (C2 sick population vs. C2 healthy population), the t-test showed a *p*-value of < 0.0001 (Figure 4); in the third dataset (X sick population vs. X healthy population parameter), the t-test showed a *p*-value of < 0.0001 (Figure 5); in the fourth dataset, correlating the BMD value (DEXA) with the C2 value of the sick population alone, the Welch–Satterthwaite test showed a *p* value of 0.0042 (Figure 6).

## 4. Discussion

Many are the anatomical districts involved in OI, and many are the signs and symptoms, including bone fragility. The patient is treated clinically and predominantly for the complex picture that may arise related to the osteoporotic condition. The dental part is often treated in second-order and is generally limited to the treatment of DI. The observation of the characteristic facies of the patient means that, today, with the new methodologies, it is possible to deepen the single components of the anatomical alterations of the maxillofacial district which can be indicators of the severity of the disease. The dentoalveolar and craniofacial anomalies, detectable by imaging, are present in all types of OI, from mild to moderate to severe (Figure 7 and Figure 8). The facial appearance of patients with both moderate and severe OI is often characterized by the triangular shape of the skull, sometimes macrocephaly, the protrusion of both temporal bones, and the prominence of the frontal bone. In addition, they have a dolic-type facial biotype, with increased vertical diameters.

Many authors [26], even today, in an attempt to understand what are the characteristic malformations of the pathology and in an attempt to find solutions, have dwelt on the malocclusion pictures [27,28,29], but have not deepened the morphological variation related to the mandible and the chin region. Instead, our study has, as a primary objective, searched for new correlation indices between anatomical variations of the chin and alveolar bone symphysis and the severity of the disease in order to find more defined anatomical details.

An in-depth study of the stomatognathic system and occlusal situation of patients with OI is essential in order to implement predictable and complication-free dental and possibly orthodontic therapy. It is also essential to intercept early, in the developmental age, growth problems of the dental arches and jaws, which can be managed with appropriate orthodontic devices, which can also be used for such patients, to eliminate or at least reduce a more severe malocclusion picture in future adult patients. Therefore, first of all, it is advisable for children with OI to have their first dental check-up early, to carry out periodic checks, and invest in prevention to avoid the onset of severe complications in adulthood.

Through level II radiological examinations, such as CBCT, it is possible to make a correct diagnosis, plan treatment with anatomical details as precise as possible, and conduct ongoing evaluations of the orthodontic intervention. Intercepting a possible variation in the preoperative phase each anatomical variability reduces the risk and prepares the operator for a possible second intervention. Some OI patients need to have a customized orthodontic treatment at diagnosis; the treatment requires the displacement of the dental elements and, as sometimes the extraction of these can cause an eruptive obstacle in the case of deciduous teeth, extraction may also be required for space problems of permanent elements. It is a great benefit for the patient to decrease the risk, facilitate the objectivity of treatment, and promote a clinical course without complications due to dental mobility or other predictable periodontal troubles by correctly measuring bone thickness. Planning can effectively prevent iatrogenic damage. The patient’s benefit is achieved by reducing complications and eliminating or limiting a worse malocclusion setting. Therefore, it is essential that the dentist, given the anatomical variability found in our study, pays close attention to some warning signs, probably pathognomonic of OI, and sometimes already present in necessary diagnostic investigations (orthopantomography and periapical intraoral radiography), which may already suggest an unusual anatomical situation and therefore requires further in-depth diagnostic examinations (CBCT) [30].

## 5. Conclusions

The diagnostic imaging of this district has undergone, over the last few years, a development that provides more anatomical details.

The present work has made it possible to identify some imaging parameters—i.e., the coronal amplitude and the eminence of the chin-bearing symphysis, which assist the dentist in the diagnostic decisions of patients with OI. This parameter has been analyzed thanks to the new CBCT technology. Many other points of interest could have been examined, but the anatomical region on which we wanted to focus the study is the chin-bearing symphysis.

This study shows that patients affected by OI have a characteristic chin-bearing symphysis, thinned and narrowing towards the center, which configures it with a constant “hourglass” appearance, not reported so far in the literature by any author. Moreover, in the study group of the sick population, a reduced bone volume and reduced thickness at the cortical, vestibular, and lingual cortical levels and at the anterior region of the mandible were appreciated (Figure 9).

Our study was born from the need to perform the second phase of orthodontic treatment, with fixed multibrackets, aimed at the finalization of the case, in patients with OI already undergoing the first phase of interceptive orthodontic therapy. From our examination, several clinical recommendations emerge, crucial for the diagnostic phase, to intercept preoperatively any anatomical variability present (for example, the “hourglass” shape of the jaw) and consequently plan the dental treatment in patients with osteogenesis imperfecta to minimize complications.

The study has focused on correlating the quantity and quality of bone by providing objective guidance for a more correct choice of orthodontic treatment and limits the risks of the treatment itself (if there is poor alveolar bone, the teeth move but they do not anchor because there is no bone); such planning with CBCT also helps follow up.

The following are therefore essential: an adequate knowledge of anatomy, a careful anamnestic evaluation, a complete radiological evaluation of the patient in comparison with the real anatomy— through specific and thorough diagnostic examinations (e.g., CBCT)—and planning of orthodontic treatment as accurately as possible, to manage complications rationally and effectively, should they arise.

## Figures and Tables

**Figure 1 healthcare-08-00546-f001:**
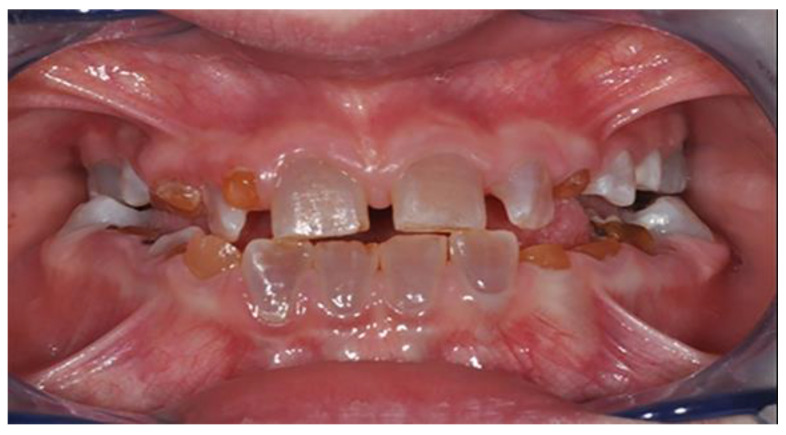
Imperfect dentinogenesis: the dental elements show a color ranging from opalescent grey to yellow.

**Figure 2 healthcare-08-00546-f002:**
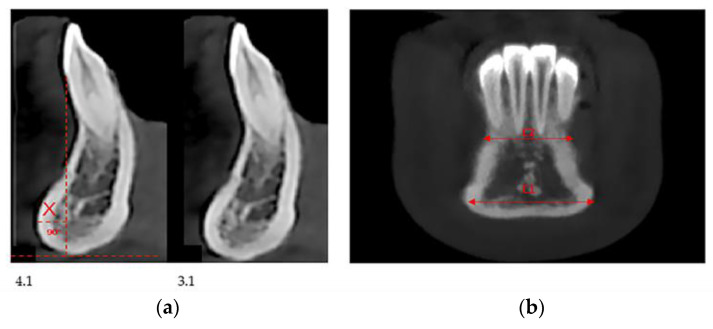
Cone beam Computed Tomography (CBCT) measurements in the paraxial 4.1, 3.1. measurement X-plane (**a**) and coronal measurement C2 and C1 (**b**) plane.

**Figure 3 healthcare-08-00546-f003:**
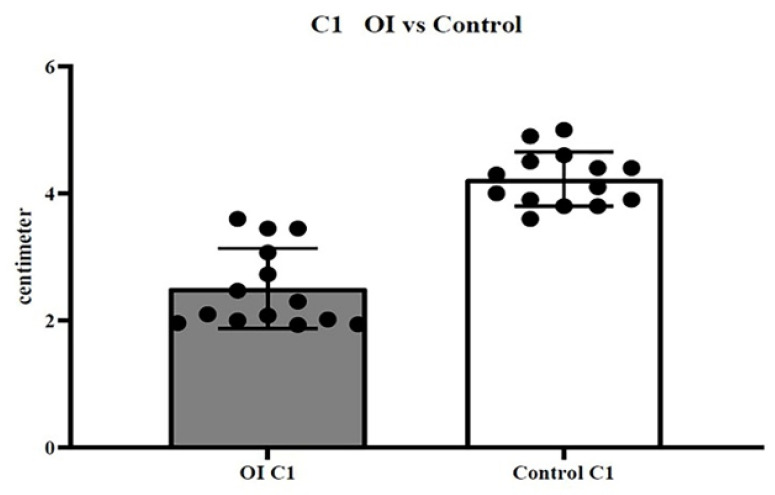
First dataset (C1 sick population vs. C1 healthy population); *t*-test showed a *p*-value < 0.0001.

**Figure 4 healthcare-08-00546-f004:**
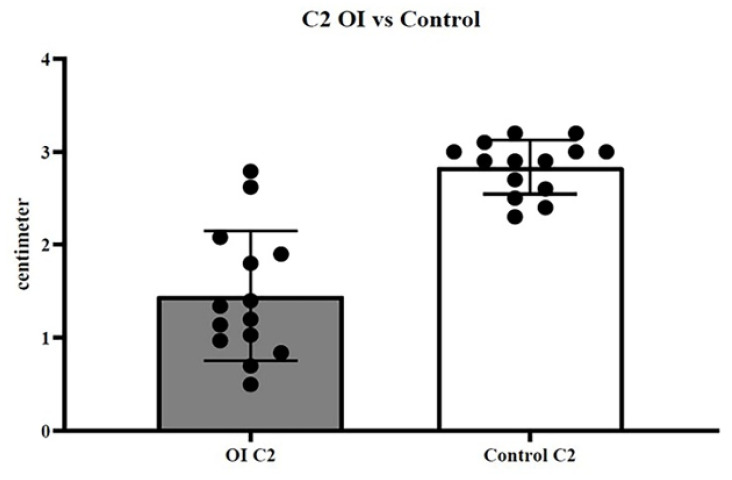
Second dataset (C2 sick population vs. C2 healthy population); the *t*-test showed a *p*-value < 0.0001.

**Figure 5 healthcare-08-00546-f005:**
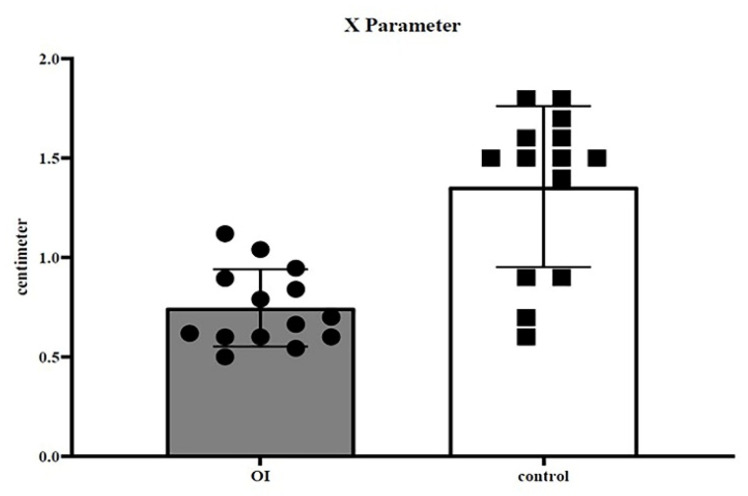
Third dataset (parameter X of the OI population vs. parameter X of the control group population); the *t*-test showed a *p*-value < 0.0001.

**Figure 6 healthcare-08-00546-f006:**
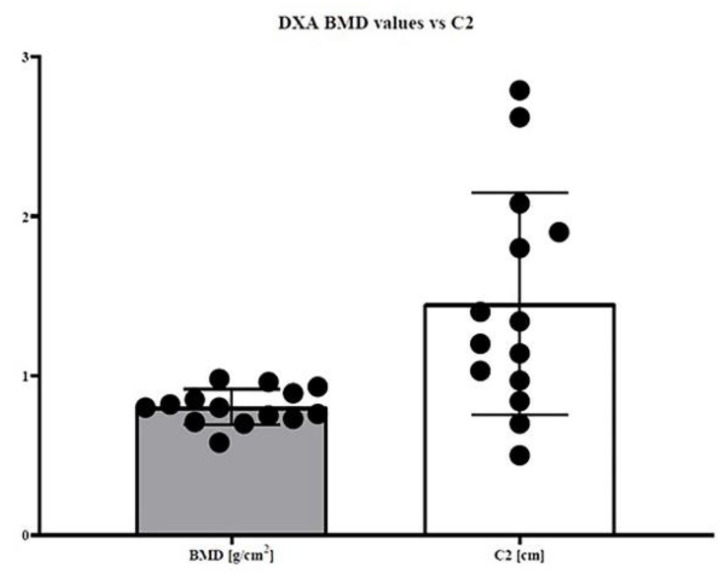
Fourth dataset (the BMD value measured using the Double Energy X-ray Absorptiometry (DEXA) technique compared to the C2 value of the population affected by OI alone) in the Welch–Satterthwaite *p*-value of 0.0042.

**Figure 7 healthcare-08-00546-f007:**
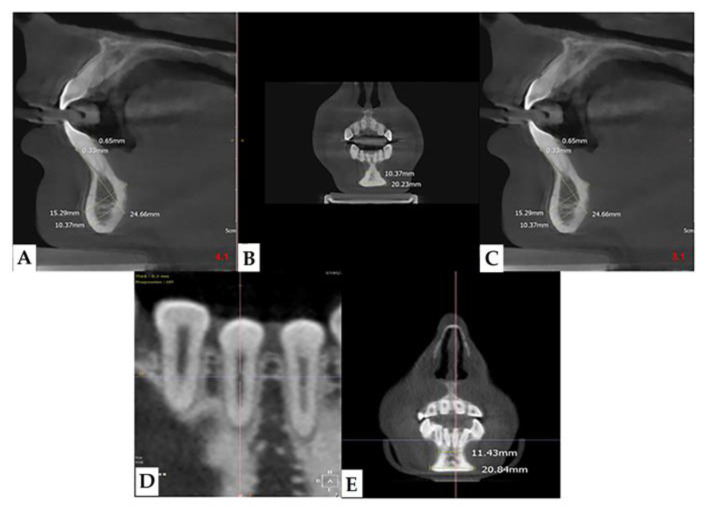
CBCT images showing bone resorption in the mandible anterior region in a patient with mild OI (**A**), sagittal 4.1, (**B**) coronal, (**C**) sagittal 3.1, (**D**), and (**E**) particular.

**Figure 8 healthcare-08-00546-f008:**
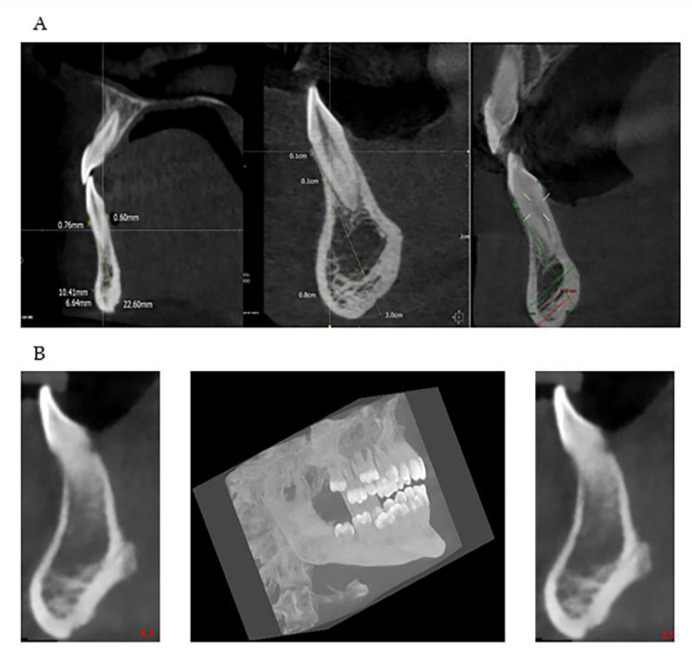
(**A**,**B**): Vestibular and lingual cortical reduction in zones 3.1 and 4.1 in patients with OI in the moderate-severe form. In the lower line of the Images (**B**) in the middle, there is a three-dimensional CBCT reconstruction that shows the very evident chin typical of OI.

**Figure 9 healthcare-08-00546-f009:**
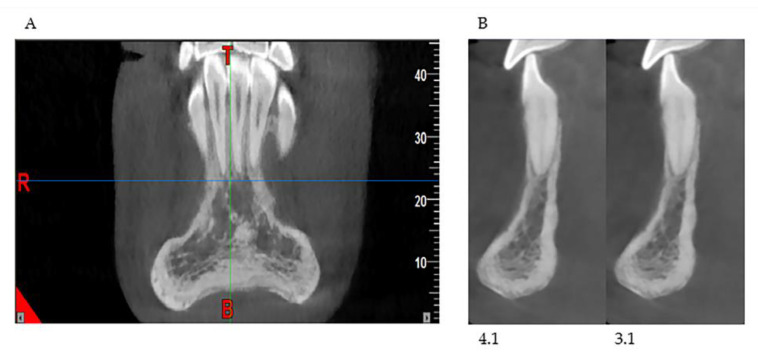
Hourglass-looking chin in the coronal reconstruction (**A**); (**B**) the paraxial image shows the accentuated chin protuberance at the lower end of the symphysis, with thinned cortical.

**Table 1 healthcare-08-00546-t001:** Patient Control Population.

Patient Code Number	Age	C1 *	C2 **
BA	14	3.0	1.6
MA_1	16	2.6	0.9
MB	13	2.9	0.7
NL	14	3.1	0.6
FM	16	2.4	0.9
TG	16	3.0	1.8
LG	16	3.2	1.5
RM	16	2.3	1.8
DSP	17	2.5	1.4
VF	12	2.9	1.6
DLA	11	3.0	1.5
MA_2	14	2.7	1.5
MA_3	16	2.9	1.7
CIR	16	3.2	1.5

* C1: Rest Chin Symphysis coronal superior line minimum average (cm); ** C2: Rest Chin Symphysis coronal lower line maximum average (cm).

**Table 2 healthcare-08-00546-t002:** Patient Population.

Patient Code Number	Age	DoD	C1 *	C2 **	BMD (g/cm^2^)
SMC	17	a	1.143	2.084	0.726
RA	13	a	2.080	3.079	0.800
OL	14	a	1.037	2.023	0.927
MM	16	a	1.2	2.0	0.983
RD	16	a	1.4	2.1	0.766
AA	17	a	1.8	2.3	0.892
AA	13	a	0.5	1.93	0.710
NT	13	b	1.344	2.473	0.745
PS	16	b	2.625	3.452	0.853
PA	14	b	0.977	1.964	0.803
LC	15	b	0.84	2.73	0.960
DSF	14	b	1.9	3.6	0.581
NM	18	b	0.7	1.95	0.823
BL	15	c	2.79	3.45	0.700

* C1: Rest Chin Symphysis coronal superior line minimum average (cm); ** C2: Rest Chin Symphysis coronal lower line maximum average (cm); DoD: degree of the disease a,b,c, Patients with Osteogenesis Imperfecta (OI) in different clinical forms, from mild (a) to moderate (b) to severe (c); BMD: bone mineral density.

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
