# Peer review of "New 3D Cone Beam CT Imaging Parameters to Assist the Dentist in Treating Patients with Osteogenesis Imperfecta"

_healthcare, 2020, doi:10.3390/healthcare8040546_

Round 1

Reviewer 1 Report

This paper reports the findings of an approach to help provide data to improve the qualities of care for patients who are suffering from osteogenesis imperfecta. The study using data from 28 patients report the findings from 4 analyses undertaken all with statistical significance which conclude that the patients from which the data were gathered have certain clinical findings derived from the clinical methodology used including “..chin-bearing symphysis, thinned, and narrowed towards the centre, configuring it with a constant "hourglass" appearance,”.

The problem with the paper at present is that there are no indication of what the initial problem is from either the clinicians or perhaps more importantly the patients perspective or how these data will address these through the additional knowledge.

The introduction provides a detailed coverage of osteogenesis imperfecta but what would be more appropriate are the shortcomings of current treatment modalities and how the additional information (the modified clinical care regime) would help address them.

Furthermore, the data may show some statistical significance but whether they are of clinical significance is the critical factor. At the end of the course of treatment are patients ‘better off’ having undergone the new approach when compared to existing care regimes.

Indeed, given the statement that the control population was ‘free from osteoporotic pathologies (lines 122 and 123)’, I might expect to see some of the reported findings. What the authors need to undertake a study comparing two groups with pathologies in which the control group is formed by those following current accepted ‘best practice’. The outcomes from the patients and/or clinicians perspectives should be reported.

Author Response

Authors Responses:

This paper reports the findings of an approach to help provide data to improve the qualities of care for patients who are suffering from osteogenesis imperfecta. The study using data from 28 patients report the findings from 4 analyses undertaken all with statistical significance which conclude that the patients from which the data were gathered have certain clinical findings derived from the clinical methodology used including “..chin-bearing symphysis, thinned, and narrowed towards the centre, configuring it with a constant "hourglass" appearance,”.

The problem with the paper at present is that there is no indication of what the initial problem is from either the clinicians or perhaps more importantly the patient’s perspective or how these data will address these through the additional knowledge.

  • We insert line 136-139‘’ This method of study is the first of its kind there are no similar studies in the literature no one had ever noticed the possibility of correlating the physiognomic variation of the chin with an objective parameter that can relate them to the clinical severity of the disease.

The introduction provides a detailed coverage of osteogenesis imperfecta but what would be more appropriate are the shortcomings of current treatment modalities and how the additional information (the modified clinical care regime) would help address them.

  • We insert line 299-302 The study has focused on correlating the quantity and quality of bone by providing objective guidance for a more correct choice of orthodontic treatment and limits the risks of the treatment itself (if there is poor alveolar bone - you move the teeth but they do not anchor because there is no bone) such planning with CBCT also helps follow up.

Furthermore, the data may show some statistical significance but whether they are of clinical significance is the critical factor. At the end of the course of treatment are patients ‘better off’ having undergone the new approach when compared to existing care regimes.

  • -We insert line 270-274 Through level II radiological examinations, such as CBCT, it is possible to make a correct diagnosis, treatment planning with anatomical details as precise as possible and ongoing evaluation of the orthodontic intervention. Decrease the risk, facilitate the objectivity of the treatment, and promote a clinical course without complications due to dental mobility or other predictable periodontal problems by correctly measuring bone thickness.

Indeed, given the statement that the control population was ‘free from osteoporotic pathologies (lines 122 and 123)’, I might expect to see some of the reported findings. What the authors need to undertake a study comparing two groups with pathologies in which the control group is formed by those following current accepted ‘best practice’. The outcomes from the patients and/or clinicians perspectives should be reported.

  • This is a great starting point for a new job. Thank you. In fact, this evaluation has been reserved to an increase of the cases and to a future evaluation because for the osteoporosis evaluation in OI should also take into account all the other districts and to correlate them not only with the unaffected but also with other diseases of which they may be carriers.

Reviewer 2 Report

The authors performed this cohort study in order to identify some characteristics that can assist the dentist in the diagnostic evaluation of craniofacial alteration and can improve orthodontic treatment of patients affected by of Osteogenesis Imperfecta.

The study is nicely written, it presents novelty merits and provides new and original information about Osteogenesis Imperfecta disease.

Authors reported that “The patient population underwent CBCT investigation, required to evaluate the second phase of orthodontic therapy, concerning a possible treatment of fixed multibracket orthodontics”

If patients underwent a mandibular CBCT for orthodontic diagnosis purpose I assume patients underwent also lateral cephalograms for a comprehensive orthodontic diagnostic evaluation.

It would be interesting if authors could provide more information about the sample providing the following information:

Sagittal skeletal discrepancy reported with ANB angle

Vertical skeletal pattern reported with the amount of divergence of mandibular plane.

It is known that the vertical skeletal pattern is able to affect the morphology of mandibular symphysis and the thickness of cortical mandibular bone.

With more information about the sample, results interpretation would be, in my opinion, more interesting for the reader.

Author Response

Dear Reviewer, we have implemented all the recommendations to optimize the paper. Sorry but I have not copied all the suggested points. However, we have taken the evaluation into account to optimize the work.

Thanks

Reviewer 3 Report

Authors reported 3D cone-beam CT new imaging parameters to assist the dentist in patients with Osteogenesis Imperfecta. The applicability of the work reported in this manuscript was required for orthodontic treatment. However, authors need to improve their manuscript using the following comments before recommending this article for publication in the journal of healthcare.

  1. Please provide the expansion of CBCT and DEXA.
  2. The quality of the Figure. 3 - 6 is very poor. Authors should revise
  3. The authors did not provide the axis parameter in Figure 6. The authors should correct it.
  4. What does mean line no 271?
  5. This article contains multiple scientific terms but has some errors. For example, the author has written (Line no. 243) the term “pathognomic”? Is correct or “pathognomonic”. Therefore, the authors are requested to double-check all.
  6. Table 1 & 2, the patient column mentioned BA, MA, …. What is that? Patient name? Make it clear.
  7. Table 2 can be separated by the order of mild (a) to moderate (b) to severe (c) without mixing them. Which is easy to correlate and understand the conditions. Please provide the expansion of BMD in Table 2 parenthesis.
  8. What basis did the author fix the condition of CBCT? What is the total time for one patient? Is there any possibility of optimizing any condition and reducing total time?
  9. Author should also discuss the disadvantage of CBCT and DEXA in OI observation in the conclusion.
  10. Author should care about grammatical errors. For example (Line no. 260) “already previously”

Author Response

  1. Please provide the expansion of CBCT and DEXA: introduction paragraph we have inserted from line 104 to line 120
  2. The quality of the Figure. 3 - 6 is very poor. Authors should revise: we have implemented 300 dpi image resolution
  3. The authors did not provide the axis parameter in Figure 6. The authors should correct it: we have entered the parameters of the x-axis in the figure
  4. What does mean line no 271? It is in the form and we have deleted it as the consent of the data to the processing and to line 147 of the text ''All patients signed written informed consent, and the study was approved by the Institutional Review Board of territorial NHS facilities (n. 260919)''
  5. This article contains multiple scientific terms but has some errors. For example, the author has written (Line no. 243) the term “pathognomic”? Is correct or “pathognomonic”. Therefore, the authors are requested to double-check all. Thank you we have corrected the terminologies in particular the typing error -Line no. 243
  6. Table 1 & 2, the patient column mentioned BA, MA, …. What is that? Patient name? Make it clear: We have specified that it is the patient code:
  7. Table 2 can be separated by the order of mild (a) to moderate (b) to severe (c) without mixing them. Which is easy to correlate and understand the conditions. Please provide the expansion of BMD in Table 2 parenthesis: we have rearranged in relation to the clinical severity the table and explained BMD in the parenthesis.
  8. What basis did the author fix the condition of CBCT? What is the total time for one patient? Is there any possibility of optimizing any condition and reducing total time? The time is related to the equipment used. By optimizing the CBCT we may have lower exposure doses and times.
  9. Author should also discuss the disadvantage of CBCT and DEXA in OI observation in the conclusion. We opted to clarify in the introduction the use and disadvantages of the methods. In our study the data are preliminary to have a significance on methods now approved in the scientific world to monitor the OI disease.
  10. Author should care about grammatical errors. For example (Line no. 260) “already previously”: done

Round 2

Reviewer 1 Report

This paper is a resubmission of an earlier version in which a number of aspects concerning the justification for and reporting of the findings were weak. Overall the authors have made a number of changes which substantially improves the basis for and justification for their comments.

Should the authors be able to the paper could have increased benefit by providing more detail in exactly how these findings will benefit patients as the current additional wording is incomplete (lines 273 -275).  

Author Response

Dear Reviewer

We have inserted Line 273-285 to clarify what is required. We hope to have clarified what the benefits for patients are.  Perhaps in the future we could think about objectifying the patient's benefit with some individual questionnaire. What has been holding us back is the age of the patients. Other studies that objectify improvements only with cephalometrics are also under consideration, but this is the first study with different parameters and using CBCT.
